# Tailoring Mesopores and Nitrogen Groups of Carbon Nanofibers for Polysulfide Entrapment in Lithium–Sulfur Batteries

**DOI:** 10.3390/polym14071342

**Published:** 2022-03-25

**Authors:** Snatika Sarkar, Jong Sung Won, Meichun An, Rui Zhang, Jin Hong Lee, Seung Goo Lee, Yong Lak Joo

**Affiliations:** 1Robert Frederick Smith School of Chemical and Biomolecular Engineering, Cornell University, Ithaca, NY 14853, USA; ss@conamix.com (S.S.); jw2636@cornell.edu (J.S.W.); ma2242@cornell.edu (M.A.); rz299@cornell.edu (R.Z.); jl3672@cornell.edu (J.H.L.); 2Department of Advanced Organic Materials & Textile Engineering, Chungnam National University, Daejeon 34134, Korea; lsgoo@cnu.ac.kr

**Keywords:** Lithium–Sulfur batteries, mesoporous carbon nanofiber, nitrogen doping, gas assisted electrospinning, air-controlled electrospray

## Abstract

In the current work, we combined different physical and chemical modifications of carbon nanofibers through the creation of micro-, meso-, and macro-pores as well as the incorporation of nitrogen groups in cyclic polyacrylonitrile (CPAN) using gas-assisted electrospinning and air-controlled electrospray processes. We incorporated them into electrode and interlayer in Li–Sulfur batteries. First, we controlled pore size and distributions in mesoporous carbon fibers (mpCNF) via adding polymethyl methacrylate as a sacrificial polymer to the polyacrylonitrile carbon precursor, followed by varying activation conditions. Secondly, nitrogen groups were introduced via cyclization of PAN on mesoporous carbon nanofibers (mpCPAN). We compared the synergistic effects of all these features in cathode substrate and interlayer on the performance Li–Sulfur batteries and used various characterization tools to understand them. Our results revealed that coating CPAN on both mesoporous carbon cathode and interlayer greatly enhanced the rate capability and capacity retention, leading to the capacity of 1000 mAh/g at 2 C and 1200 mAh/g at 0.5 C with the capability retention of 88% after 100 cycles. The presence of nitrogen groups and mesopores in both cathodes and interlayers resulted in more effective polysulfide confinement and also show more promise for higher loading systems.

## 1. Introduction

As demand for energy increases, there is a growing need for clean, efficient, and cost-effective renewable energy sources and to reduce the usage of fossil fuels. Lithium–Sulfur (Li–S) batteries have a high theoretical energy density of 2600 Wh/kg and a specific capacity of 1675 mAh/g. Sulfur is abundantly available and is environmentally benign [1,2,3]. All these reasons make Li–S batteries among one of the most promising next-generation rechargeable batteries.

However, Li–S batteries face several challenges in performance that need to be resolved before they can become commercially viable. One of these challenges is the dissolution of higher order polysulfides into the electrolyte, which makes it difficult to recover them as lithium–sulfide precipitate on the cathode at the end of discharge cycle, leading to loss of active material and low capacity. Another challenge is the polysulfide shuttle effect, which occurs when the higher order polysulfides migrate to the anode, become reduced to Li_2_S, and migrate back to the cathode to become re-oxidized. The insulating nature of Sulfur and lithium–sulfide also adds to the cell resistance. Moreover, volume changes and the formation of lithium dendrites hinder the development of Li–S batteries [3,4,5,6].

Several efforts have been made to combat the technical issues mentioned above. It has been found that different types of carbon nanomaterials such as microporous carbon, mesoporous carbon, hierarchical porous carbon, carbon black, hollow carbon spheres, carbon nanotubes (CNTs), carbon nanofibers (CNFs), reduced graphene oxide, and graphene can improve cell performance. These porous carbons can contain the active material, constrain the dissolved polysulfides, and accelerate charge/electron transport [7,8,9,10,11,12,13]. Manthiram et al., introduced the concept of inserting an interlayer between the cathode and separator to act as a polysulfide entrapper and enhance the re-utilization of trapped material. However, despite the improved performance, this acted as a temporary solution since the interlayer acted as only a physical barrier. The polysulfides can eventually leach out to the electrolyte [14].

To ensure more permanent constraining of polysulfides, chemical modification of active the host’s surface is also effective in reducing shuttle effect and improving cycle performance. The introduction of doping elements such as nitrogen, sulfur, phosphorus, and boron have been shown to form strong chemical bonds between the polysulfides and these elements [15,16]. Nitrogen doping has been carried out by introducing electron-rich functional groups such as amines, polypyrrole, pyrroles, and pyridines. It has been found to be effective in assisting mesoporous carbon in suppressing polysulfide shuttling effect through Lewis acid–base interactions with polysulfides [17,18,19,20,21,22].

In this paper, we study and compare the results of creating mesopores in carbon nanofibers and functionalizing mesoporous carbon cathodes and interlayers with electron-rich nitrogen groups. Mesoporous carbon nanofiber webs, under various activated conditions, were synthesized by gas-assisted electrospinning of polyacrylonitrile (PAN) and polymethyl methacrylate (PMMA) blend solutions, where PMMA is used as a sacrificial polymer to create mesopores and used as both cathode and interlayer in our system. Cathodes were prepared by electrospraying sulfur onto mesoporous carbon, a facile technique that can be used to create higher loading sulfur electrodes. The surface chemistry of these mesoporous fibers was modified by introducing cyclic nitrogen groups followed by carbon dioxide activation. They were then utilized as cathodes and interlayers to test the effects of adding cyclic nitrogen groups and mesopores on cell performance.

## 2. Materials and Methods

### 2.1. Fabrication of PAN/PMMA Nanofibers

Polyacrylonitrile (PAN) (MW = 150,000 from Sigma Aldrich, St. Louis, MO, USA) and Poly (methyl methylacrylate) (PMMA) (MW = 15,000 from Sigma Aldrich, St. Louis, MO, USA) were dissolved in dimethylformamide (DMF) in separate vials by stirring for 24 h at 65 °C [23]. PAN/PMMA solutions with different blend ratios (56:44, 59:41, 77:23, and 80:20 *w*/*w*) were prepared by dissolving PAN and PMMA into DMF. In this study, we fabricated PAN/PMMA nanofibers via gas-assisted electrospinning, which is a good process to produce fiber structure by combining an electric field between the nozzle and collector, and high-speed air as driving forces. A Harvard Apparatus PHD Ultra (Harvard Apparatus, Holliston, MA, USA) was used for electrospinning the solution onto an aluminum foil current collector in a chamber controlled at 16% relative humidity. The electrospinning apparatus schematic diagram is shown in Figure 1. A coaxial needle with a 12-gauge inner needle and a 16-gauge outer shell was used. The solution was fed through the inner needle at an infusion rate of 0.03 mL/min, and 16% RH dry air was supplied through the outer shell at an air pressure of 10 psi. The tip-to-collector distance and applied voltages were 15 cm and 18 kV, respectively.

### 2.2. Synthesis of Mesoporous Carbon Nanofiber Web (mpCNF)

The spun nanofibers were peeled from the collector and dried in a vacuum oven (nXDS 10i, Edwards, Ltd., Burgess Hill, West Sussex, UK) at 80 °C for 2 h. Stabilization and carbonization of fibers were conducted in a tube furnace. The dried nanofibers heat-treated in air with a heating rate of 5 °C/min at 280 °C and held for 4 h to stabilize the PAN component of the fibers. The fibers were then placed between two ceramic plates in a nitrogen-filled tube furnace with a heating rate of 10 °C/min at 1050 °C and held for 1 h to carbonize the PAN component and remove the PMMA component by thermal degradation creating multi porous carbon nanofibers. During the activation of carbon nanofibers, the pressure load was applied from 0 to 1.28 N, as illustrated in Figure 1.

### 2.3. Synthesis of Cyclized-Polyacrylonitrile Modified CNF Fibers (CPAN/mpCPAN)

A 5 wt% PAN-PMMA solution in DMF was prepared. The above prepared CNF nanofibers were immersed in the PAN solution for 1 m and dried at 80 °C for 12 h. The PAN-coated mesoporous fiber was heat-treated in Argon at 300 °C for 10 h to form cyclized PAN nanofibers (CPAN) [24]. The as-prepared CPAN fibers were activated in a heat-treatment process under carbon dioxide at 700 °C for 1 h to improve the mesopore distribution and BET surface area. CPAN after carbon dioxide activation is referred to as mesoporous cyclized PAN (mpCPAN).

### 2.4. Synthesis of Electrodes

The electrodes were synthesized by air-controlled electrospray. Sublimed sulfur (Spectrum Chemical Mfg. Corp., New Brunswick, NJ, USA) and Ketjen Black EC-600JD (KB) (AzkoNobel, Washington, DC, USA) solution were dispersed in CS2 to obtain a composition of 97% Sulfur and 3% Ketjen Black [25]. The solution was stirred for 3 h and sonicated for 30 min before electrospraying on mpCNF and mpCPAN webs at 15 kV, 10 cm distance from the collector at 0.08 mL/min flow rate. An 18-gauge stainless steel needle was used for air-controlled electrospraying. Electrospraying was carried out until the target Sulfur loading was deposited on the cathode. Cathodes with sulfur loading of 1.1 mg/cm^2^ were used for testing. The sulfur content of the substrates was 25.0% (14.3% including the interlayer mass) for 1.1 mg/cm^2^.

### 2.5. Material Characterization

The pore size distribution analysis was performed on a Micrometrics Gemini VII 2390 (Micromeritics Instrument Corp., Norcross, GA, USA) in liquid nitrogen with the Brunauer, Emmett, and Teller (BET) method. Samples were degassed under nitrogen at 300 °C for at least 3 h. The scanning electron microscope (SEM) images were taken by a TESCAN Mira3 Field Emission SEM and Zeiss Gemini 500 SEM (Zeiss, Oberkochen, Germany). The Fourier Transform Infrared (FTIR) spectra of mpCNF and mpCPAN were carried out using a Bruker Hyperion FT-IR Microscope (Bruker, Billeric, MA, USA) at ambient temperature. X-ray Photoelectron Spectroscopy (XPS) of mpCPAN and mpCNF before and after cycling was used to obtain chemical bonding information and elemental analysis (ESCA 2SR, Scienta Omicron, Danmarksgatan, Uppsala, Sweden).

### 2.6. Electrochemical Characterization

The interlayers and substrates were cut into discs of diameter 15 mm from the nanofiber mat. CR2032 coin cells were assembled using sulfur deposited mpCPAN/mpCNF (5.5 mg/cm^2^) as cathodes, lithium metal (same diameter as cathode) as anode, mpCPAN/mpCNF (8.2 mg/cm^2^) as interlayers, a 25-micron thick Celgard separator (2400, Celgard, LLC, Charlotte, NC, USA), an electrolyte of 1 M lithium bis(trifluoromethanesulfonyl)imide, and 0.2 M LiNO_3_ in 1,2-dioxolane/1,2-dimethoxyethane (*v*/*v* = 1:1). EIS measurements were performed using a Solartorn Cell Test System model 1470E potentiostat between frequencies 0.01–100kHz (Solartron Analytical, Leicester, UK).

## 3. Results and Discussion

### 3.1. Surface and Structural Properties of Multi-Porous Carbon Nanofibers

Carbon nanofibers were fabricated by electrospinning PAN as the precursor and PMMA as the sacrificial polymer to obtain the porous structure. The multi-porous carbon nanofiber electrode materials are obtained by the carbonization-activation step. The pore size could be controlled by simultaneously applying the loading pressure in the carbonization-activation step. In many published papers, pores were prepared through the weight ratio of PAN/PMMA, but it was difficult to control the size and uniformity of meso- or macro-pores. The SEM image in Figure 2 shows carbon nanofibers prepared by applying a load of 0 to 1.28 N during carbonization activation at a PAN/PMMA weight ratio of 59:41. As the applied load increased, the carbon nanofiber diameter increased from about 220 nm to 400 nm. Macropores were observed on the carbon nanofiber surface, possibly because the applied load helps to withhold shrinkage and collapse of pores created by the removal of PMMA during the carbonization-activation stage. The fiber diameter and macropore are one of the important factors determining the electrochemical properties of carbon materials formed from electro-spun nanofibers.

The pore size can also be controlled by varying the amount of PMMA in the blend compositions. Appendix A shows the morphological changes of the surface of carbon nanofibers prepared by varying the weight ratio of PAN/PMMA under a load of 1.28 N. SEM images (Appendix A) revealed that the average diameter of fibers increases with increasing PAN amount in the blend compositions. The fiber diameter obtained for PAN/PMMA (56:44) was 390 nm increasing to 440 nm for PAN/PMMA (77:23) and 450 nm for PAN/PMMA (80:20). The trend can be attributed to the lower viscosity of the solutions with higher amounts of PMMA compared to that of a neat PAN solution. During electrospinning, the dispersed PAN droplets, having a higher viscosity compared to that of PMMA, would be elongated more by the electrical forces, and thus the fiber diameter increased further. These results suggest that the larger the fiber diameter, the better the pore formation under the load pressure applied during carbonization activation. We found PAN/PMMA (80:20) to be the optimum blend composition to create meso- and macro-pores among those samples.

Figure 3 and Figure 4 show the cross-section morphology of carbonized and activated nanofibers at different pressure load and weight ratios. After the carbonization-activation treatments, the carbon nanofibers retained their fibrous morphology without collapsing. All the carbon nanofibers demonstrate various internal pore sizes. In particular, in the case of carbon nanofibers prepared at a weight ratio of PAN/PMMA (80:20) and a load of 1.28 N, mesopores and macropores were observed not only on the surface but also on the inside.

Appendix A illustrate the nitrogen adsorption–desorption isotherms and the pore size distribution for the resulting activated nanofibers, respectively. Surface area and the pore size distribution are important factors that determine the performance of the Li–S batteries. According to Table 1 and Figure 5 and Figure 6, the variables of PAN/PMMA weight ratio and loading pressure caused changes in surface area and pore size. All carbon nanofiber compositions had more macropores compared to PAN/PMMA carbon nanofiber without loading. Among carbon nanofibers, PAN/PMMA (59:41) with no load applied has the largest surface area. However, mesopores and macropores were about 14% less than those of PAN/PMMA (80:20) carbon nanofibers loaded with 1.28 N. Detailed results of the BET measurements, including specific surface area and volume percentage of total pore volume of all manufactured carbon nanofibers, are summarized in Table 1. These results demonstrated that pore size could be tuned from the application of loading pressure during the activation process. The pore size distribution and presence of mesopores are known to be important aspects of high-performance Li–S batteries due to their high sulfur loading, better access, and release of bulky polysulfides. Thus PAN/PMMA (77:23) carbon nanofibers loaded with 1.28 N, which exhibits a decent surface area and a high fraction of mesopores, has been selected for mesoporous carbon nanofibers (mpCNF) for Li–S battery applications.

### 3.2. Characterization of CPAN and mpCPAN Nanofibers

We note that the PAN coated mpCNF was heat-treated in Argon at 300 °C for 10 h to form cyclized PAN nanofibers (CPAN). The as-prepared CPAN fibers were activated in a heat treatment process under carbon dioxide at 700 °C for 1 h to improve the mesopore distribution and BET surface area. CPAN after carbon dioxide activation is referred to mesoporous cyclized PAN (mpCPAN). Various analytical tools were used to characterize the surfaces of mpCNF, CPAN, and mpCPAN and confirm the presence of cyclic nitrogen groups in mpCPAN after synthesis. XPS survey scans of mpCNF and CPAN are presented in Figure 7a. Before the cyclization reaction for mpCNF, we observe two prominent peaks for C 1s at 285 eV and O 1s at 532 eV. After the cyclization reaction for CPAN, we see a third prominent peak for N 1s at 400 eV. However, as seen in Figure 7b, after CO_2_ activation, the intensity of the nitrogen peak is reduced for mpCPAN than CPAN, indicative of lower nitrogen content. Figure 7c,d show the deconvoluted N 1s signals for CPAN and mpCPAN, respectively. Two peaks at 398 eV and 399.9 eV represent pyridine and pyrrole, respectively, [22,23,24,25]. The ratio of pyrrole-to-pyridine in mpCPAN after CO_2_ activation appears to be higher than that for CPAN.

FTIR spectra of mpCPAN in Figure 8 shows absorbance bands at 1066, 1150, and 1195 cm^−1^ which represent C–N stretching. The peak at 1560 cm^−1^ represents the vibration of combined C=C and C=N and the peak at 1370 cm^−1^ is for C–C stretching [22,23,24,25].

SEM images of mpCNF and mpCPAN are shown in Figure 9a,b. It is observed that a significant fraction of external pores among fibers in mpCNF is covered in mpCPAN by dip-coating treatment used to synthesize mpCPAN. It appears that the fiber diameter substantially increases after the dip-coating, followed by cyclization and CO_2_ activation.

The BET surface area reduced significantly from 547.7 m^2^/g to 32 m^2^/g in CPAN. The pore-size distribution of mpCNF and CPAN are shown and compared in Figure 10. It is observed that not only large mesopores (>20 nm) are greatly reduced, but also small meso-/micro-pores (<5 nm) are removed in CPAN, by dipping PAN followed by cyclization reaction.

However, on activation of CPAN, the BET surface area was increased to 300.6 m^2^/g and a corresponding increase in mesopore volume was also observed as shown in Figure 11. This confirms that the activation process improves the surface area of the nanofibers by sacrificing the nitrogen content, which can prove beneficial for better cathode performance.

### 3.3. Applications of mpCNF, CPAN, and mpCPAN Nanofibers in Li–S Batteries

Having confirmed the cyclization of mpCNF and the increased surface area after CO_2_ activation, mpCPAN was chosen for our nitrogen-doped mesoporous carbon cathode and interlayer material. The effects of mpCNF and mpCPAN on the electrochemical performance of the cells were tested by using them as both cathodes and interlayers in the following combinations of systems:mpCNF cathode and mpCNF interlayer (mpCNF-mpCNF)mpCNF cathode and mpCPAN interlayer (mpCNF-mpCPAN)mpCPAN cathode and mpCNF interlayer (mpCPAN-mpCNF)mpCPAN cathode and mpCPAN interlayer (mpCPAN-mpCPAN)

Cell performance for systems with mpCPAN replaced by CPAN was also examined as shown in Appendix A. These systems were analyzed with S-loading of 1.1 mg/cm^2^ and 1.7 mg/cm^2^. Despite the presence of higher nitrogen content than the remaining systems, the system CPAN-CPAN had the poorest cycle performance, possibly due to the lack of micro-/meso-pores by coating PAN on mpCNF. The deconvoluted N 1s signals for CPAN as interlayer and cathode in Appendix A, respectively in the Appendix A, also implies that using CPAN as cathode results in poorer nitrogen group utilization for polysulfide capture. In Figure 12, the cycle performance and rate capability of systems utilizing mpCPAN were examined for S-loading of 1.1 mg/cm^2^. In this case, due to the lower nitrogen content of mpCPAN, the mpCNF-mpCPAN system had a lower capacity compared to its non-activated counterpart. However, the mpCPAN-mpCPAN system had the best cycle performance, leading to the capacity of 1200 mAh/g at 0.5C after 100 cycles with capacity retention of 88%. In addition, the rate capability of the mpCPAN-mpCPAN system was comparable to that of the mpCNF-CPAN system. This may be due to the presence of mesopores and nitrogen groups in both the cathode and interlayer, leading to greater polysulfide confinement. Those for high S-loading of 1.7 mg/cm^2^ are shown in Appendix A, which exhibit the similar trend. In comparison, Kalra et al. obtained a discharge capacity of 1285 mAh/g at 0.2 C after 100 cycles with 83% capability retention using a meso–microporous carbon nanofiber interlayer [26].

The plots of Electrochemical Impedance Spectroscopy (EIS) of fresh cells are shown in Figure 13. The EIS spectra of each system consist of one semicircle (Charge transfer resistance, Rct) and slope line (Warburg Resistance). We note that the mpCNF-mpCNF and mpCPAN-mpCNF systems had an additional semi-circle after discharging, corresponding to Li_2_S precipitate film [2,27]. This may be because these systems have mpCNF as an interlayer. The polysulfides become deposited between the fibers, but may not be reutilized efficiently in the following charging-discharging cycles. For fresh cells, the charge transfer resistance of cells with at least one nitrogen-doped component was lower than the reference mpCNF-mpCNF system.

High-resolution S 2p and N 1s spectra of the cathode and interlayer of each system were examined after 100 cycles and are shown in Figure 14a–d. The cells were fully discharged to 1.8 V. The carbon spectra were calibrated to 284 eV and peak fitting was done with a Shirley Background. In the S 2p spectra for each system, a sulfate peak at 166.7 eV was observed, which can be attributed to the reaction of the polysulfides with air/moisture on exposure [28]. An additional signal was observed at 158 eV which is attributed to trapped lithium polysulfides [28,29,30,31]. To further confirm the effectiveness of the nitrogen groups in capturing polysulfides, high-resolution N 1s spectra of systems utilizing mpCPAN as cathode or interlayer or both were examined in Figure 14b,d. In the case of mpCPAN, the N 1s spectra of the interlayer were deconvoluted to obtain two peaks at 399.2 and 397 eV. Comparing this to the N 1s spectra of pristine mpCPAN, we find that the peaks corresponding to pyrollic- and pyridinic-nitrogen shifted towards a lower binding energy as a result of Li-N interaction [15,32,33]. During Li–N interaction, there is an increased electron density around the more electronegative N atom [15]. The resulting increased shielding effect can reduce the energy required to knock off the electrons.

Although mpCNF-mpCNF has both components with a higher surface area, it has a lower capacity and stability compared to mpCPAN-mpCNF and mpCNF-mpCPAN. This may be attributed to the absence of nitrogen groups to trap the polysulfides through chemical bonds. The mpCNF-mpCNF interlayer physically traps the polysulfides in the pores, which may eventually leach out into the electrolyte during discharge, or may also not get completely released for conversion to sulfur during charging (hence, it is a poor sulfur re-utilization).

Conversely, mpCPAN systems had a relatively lower N-content but increased mesopores, therefore allowing them to function effectively as both cathodes and interlayers. However, due to the reduced nitrogen content of mpCPAN, the system mpCPAN-mpCPAN has the most superior performance, comparable to that of the mpCNF-CPAN system. The presence of nitrogen groups and mesopores in both cathodes and interlayers results in more effective polysulfide confinement. It also shows more promise for higher loading systems.

## 4. Conclusions

In summary, we have combined different physical and chemical modifications of carbon nanofibers through the creation of micro-, meso-, and macro-pores as well as the incorporation of nitrogen groups in cyclic poly-acrylonitrile (CPAN) using gas-assisted electrospinning and air-controlled electrospray processes. We have incorporated them into electrode and interlayer in Li–Sulfur batteries. We have discussed the effects of using mesoporous carbon as a host for cyclic nitrogen groups on battery performance. Four systems utilizing mesoporous cyclized PAN fibers as cathode/interlayer were compared. Despite lowered nitrogen content, the increased mesopores showed a significant improvement in cell performance for all the nitrogen-containing systems, particularly mpCPAN-mpCPAN (a capacity of 1177 mAh/g at 0.5 C after 100 cycles with 88% capability retention), relative to the reference mpCNF-mpCNF system (a capacity of 897 mAh/g at 0.5 C after 100 cycles). The heat treatment of cyclized PAN fibers produced by conventional means is a facile approach to optimize both N-content as well as surface area and mesopore content, which are crucial for cathode performance. The presence of nitrogen groups and mesopores in both cathodes and interlayers results in more effective polysulfide confinement and also shows more promise for higher loading systems.

## Figures and Tables

**Figure 1 polymers-14-01342-f001:**
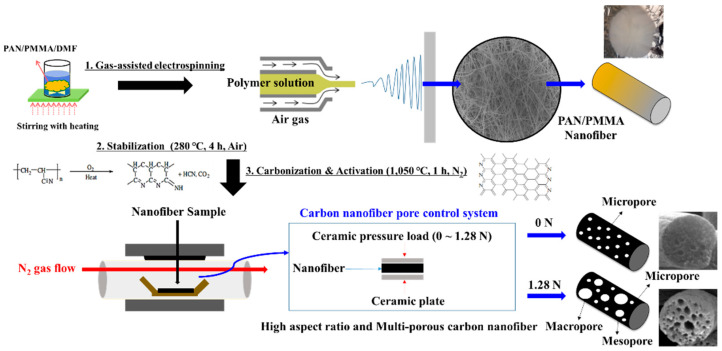
The schematic and photo images of multi-porous carbon nanofibers via ceramic plate load control.

**Figure 2 polymers-14-01342-f002:**
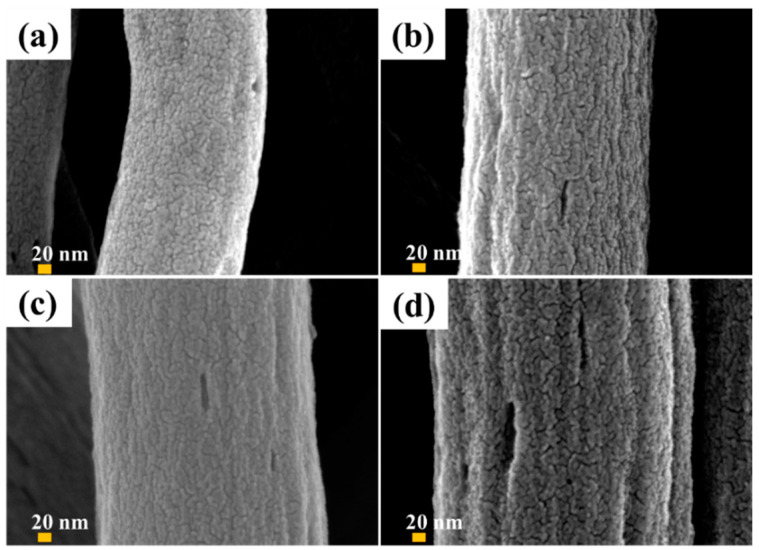
SEM surface images of carbon nanofibers obtained with different ceramic pressure load at 59:41 weight ratio of PAN/PMMA: (**a**) 0 N, (**b**) 0.62 N, (**c**) 1.12 N, (**d**) 1.28 N.

**Figure 3 polymers-14-01342-f003:**
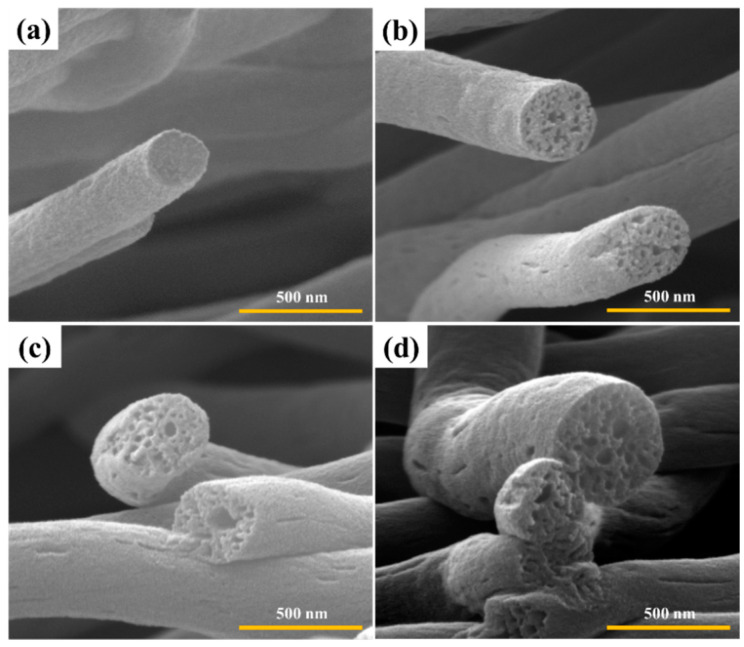
The cross-section SEM images of carbon nanofibers obtained with different ceramic pressure load at 59:41 weight ratio of PAN/PMMA: (**a**) 0 N, (**b**) 0.62 N, (**c**) 1.12 N, (**d**) 1.28 N.

**Figure 4 polymers-14-01342-f004:**
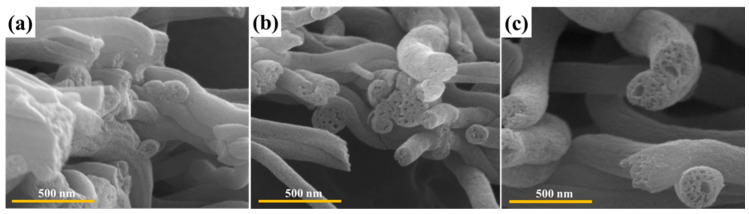
The cross-section SEM images of carbon nanofibers obtained with different weight ratio at ceramic pressure load of 1.28 N: (**a**) PAN/PMMA (56:44), (**b**) PAN/PMMA (77:23), (**c**) PAN/PMMA (80:20).

**Figure 5 polymers-14-01342-f005:**
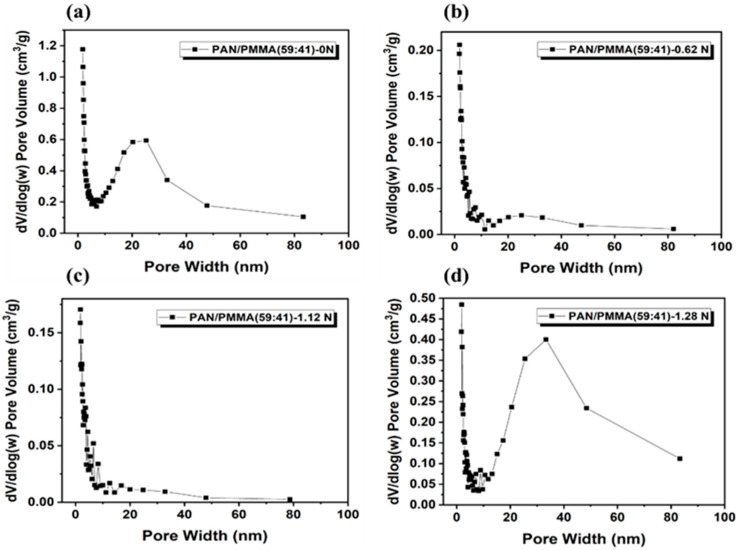
BJH analysis of nitrogen physisorption showing the pore volume distribution with ceramic pressure load: (**a**) 0 N; (**b**) 0.62 N; (**c**) 1.12 N; (**d**) 1.28 N.

**Figure 6 polymers-14-01342-f006:**
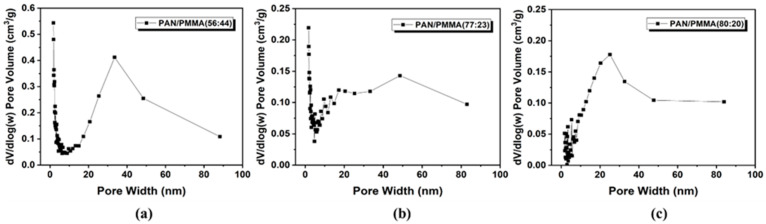
BJH analysis of nitrogen physisorption showing the pore volume distribution with PAN/PMMA weight ratios: (**a**) 56:44; (**b**) 77:23; (**c**) 80:20.

**Figure 7 polymers-14-01342-f007:**
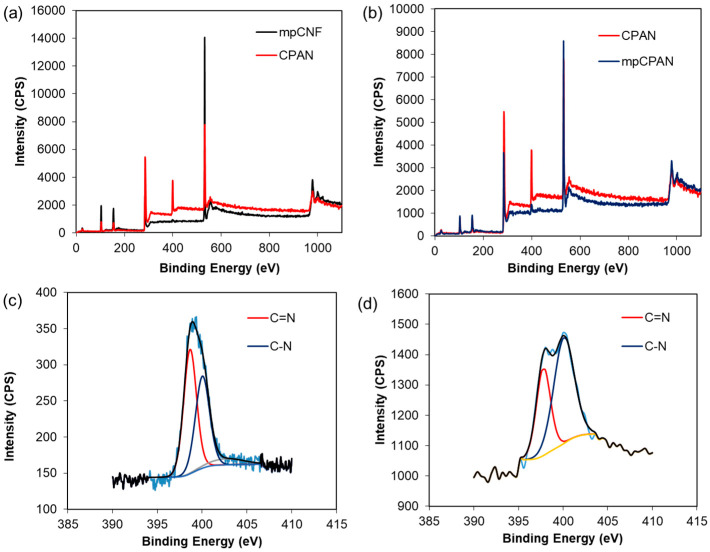
(**a**) XPS survey scanning spectra for mpCNF and CPAN; (**b**) XPS survey scanning spectra for mpCPAN and CPAN; (**c**) High Resolution N 1s spectra of CPAN; (**d**) High Resolution N 1s spectra of mpCPAN.

**Figure 8 polymers-14-01342-f008:**
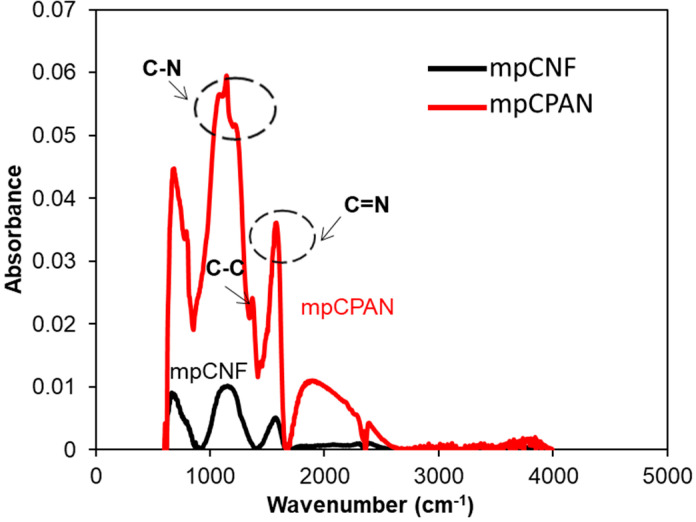
Experimental FTIR spectra of mpCNF and mpCPAN.

**Figure 9 polymers-14-01342-f009:**
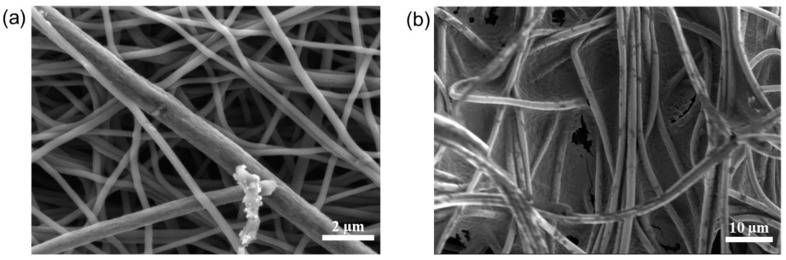
SEM images of (**a**) mpCNF and (**b**) mpCPAN.

**Figure 10 polymers-14-01342-f010:**
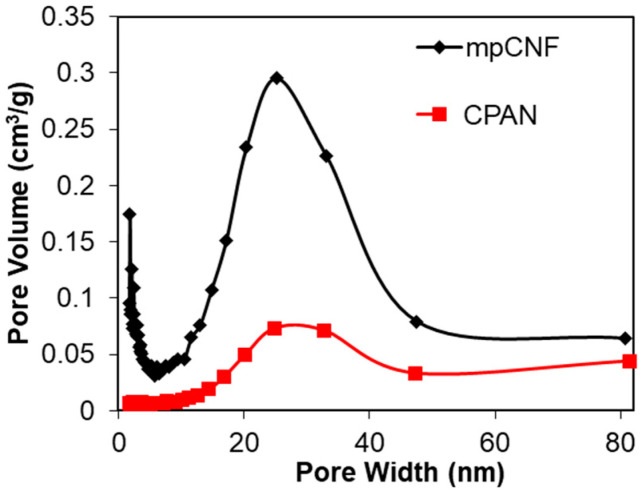
Comparison of pore-size distribution of mpCNF and CPAN assessed by the BET method.

**Figure 11 polymers-14-01342-f011:**
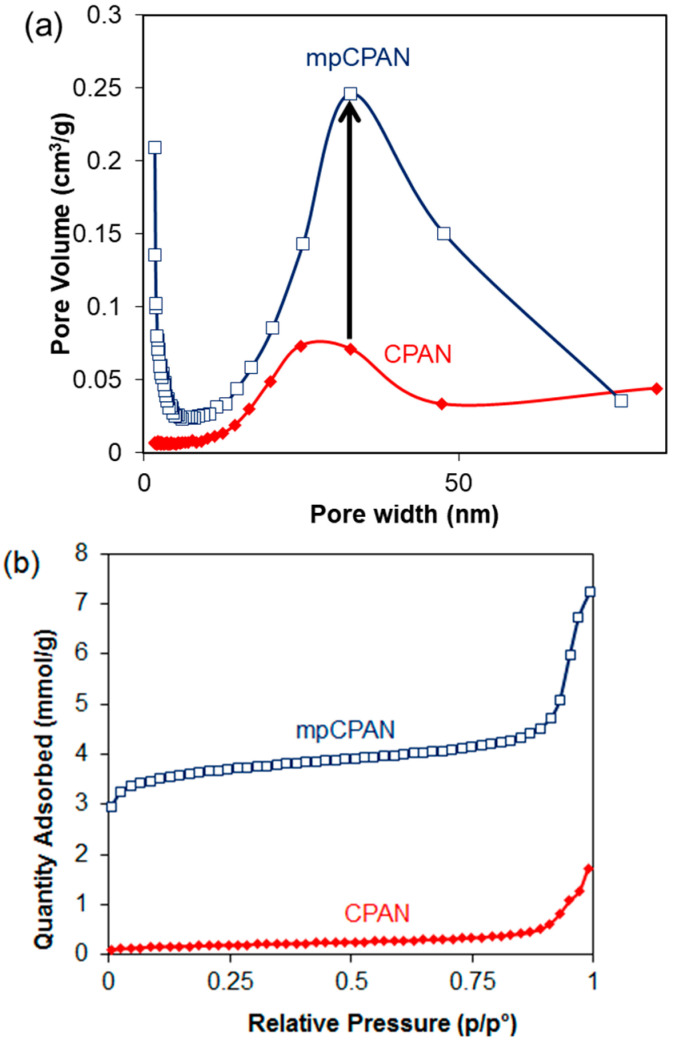
(**a**) Pore-size distribution of CPAN and mpCPAN obtained by the BET surface area analysis; (**b**) Nitrogen adsorption isotherm for CPAN and mpCPAN.

**Figure 12 polymers-14-01342-f012:**
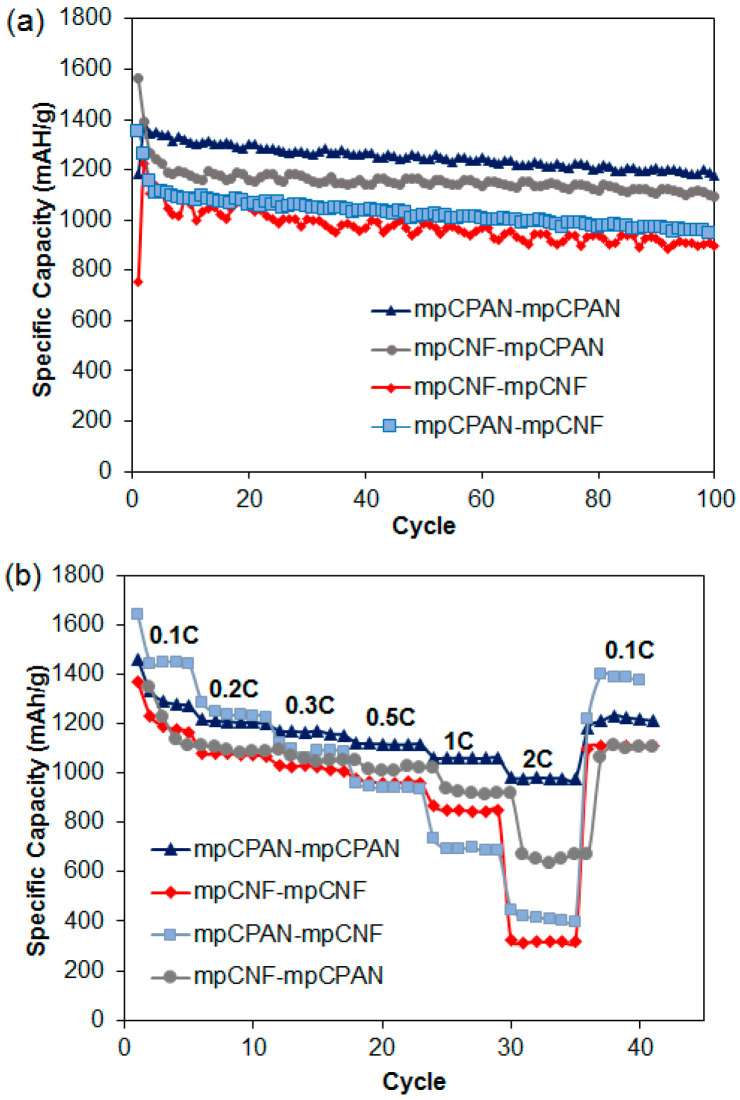
(**a**) Cycling performance of four systems at 1.1 mg/cm^2^ S-loading, 0.5 C; (**b**) Rate capability test comparison of systems using mpCPAN as interlayer/cathodes at 1.1 mg/cm^2^ S-loading.

**Figure 13 polymers-14-01342-f013:**
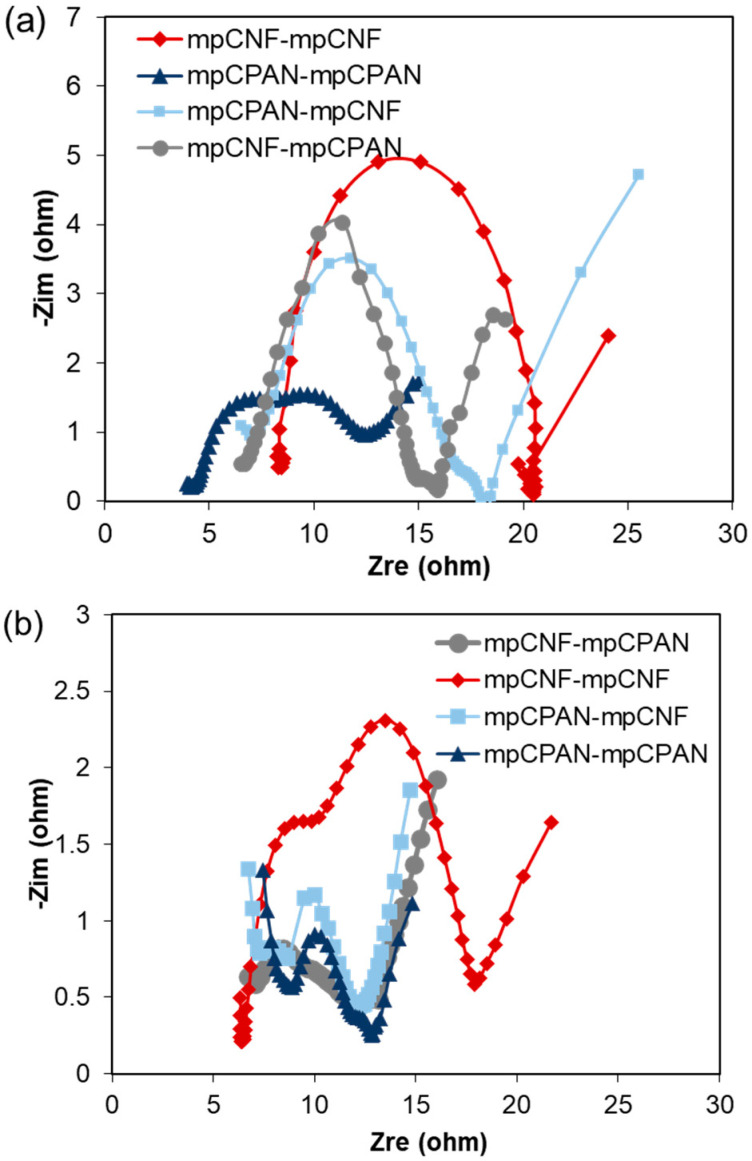
(**a**) EIS Spectra of fresh cells; (**b**) EIS Spectra of discharged cells after 100 cycles.

**Figure 14 polymers-14-01342-f014:**
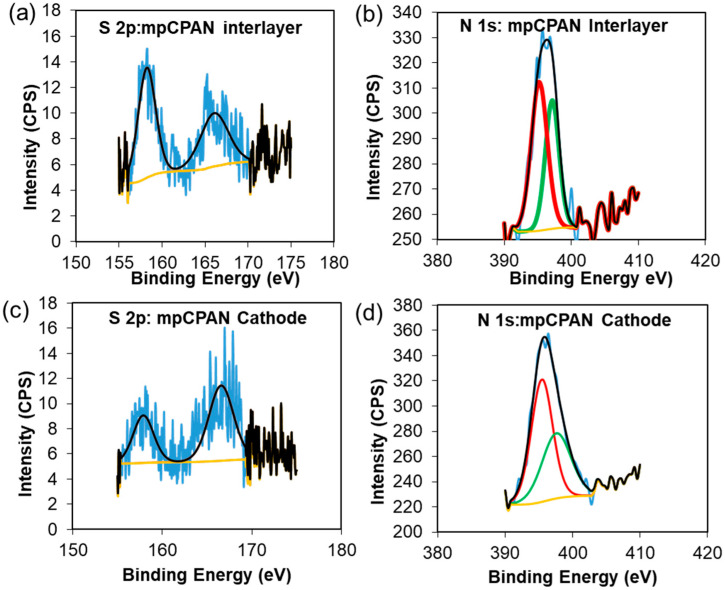
(**a**) High Resolution S 2p spectra of mpCPAN interlayer; (**b**) High Resolution N 1s spectra of mpCPAN interlayer; (**c**) High Resolution S 2p spectra of mpCPAN cathode; (**d**) High Resolution N 1s spectra of mpCPAN cathode.

**Table 1 polymers-14-01342-t001:** Textural properties of obtained carbon nanofiber materials.

Multi-Porous Carbon Nanofibers		BET Surface Area (m^2^/g)	Mesopore Volume/Total Volume (cm^3^/g)
PAN/PMMA (59:41)	0 N	1284	0.80
	0.62 N	376	0.77
	1.12 N	224	0.80
	1.28 N	420	0.81
Ceramic pressure load 1.28N	PAN/PMMA (56:44)	456	0.79
	PAN/PMMA (77:23)	547	0.84
	PAN/PMMA (80:20)	126	0.89

## Data Availability

The data presented in this study are available in the article.

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
