# Peer review of "Tailoring Mesopores and Nitrogen Groups of Carbon Nanofibers for Polysulfide Entrapment in Lithium–Sulfur Batteries"

_polymers, 2022, doi:10.3390/polym14071342_

Round 1
Reviewer 1 Report
The authors employed a holistic approach to inhibit the shuttle effect in Li-S batteries and improve cyclability and rate capability. The creation of mesopores and incorporation of N-groups onto both cathode and interlayer leads to the impressive performance of 1000 mAhg-1 at 2C and capacity retention close to 88 % after 100 cycles. The results are promising. I would recommend for publication after minor revision.
- Line 34-41 describes the challenges in Li-S batteries. You may include volume changes and Li-dendrite formation as well although the present study does not address these issues.
2. Line 118-119: The % of Sulfur is only 14.3 %. What are the thickness and total mass of the electrode?
3. Line 146-147: Why should "applied load" increase the diameter of CNF?
4. CO2 should be replaced with CO2 throughout the manuscript.
5. Conclusion, Line 335: Include specific numbers for comparison instead of just mentioning "significant improvement".
Reviewer 2 Report
In the manuscript entitled “Tailoring Mesopores and Nitrogen Groups of Carbon Nanofibers for Polysulfide Entrapment in Lithium-Sulfur Batteries“ Authors have proposed a micro-, meso- and macro-porous nanofibers with nitrogen groups as cathode greatly enhances the rate capability and capacity retention.
In my opinion, some parts of the paper should be improved:
1) Scale bar for Figure 3, Figure 4, and Figure S1 is too small, I propose to add a new scale bar without a black stripe at the bottom with additional analysis information. Also from Figure 2 a black stripe at the bottom should be removed.
2) Determined in Table 1 percentage of micropores, mesopores, and macrospores is incorrect. Because used Gemini VII aperture is dedicated for measurement only BET and measurement at higher p/p, therefore we cannot determine micropore volume with this aperture. Using the BJH method we can determine only mesopore volume between in the range of 2-50 nm, therefore I suggest removing a percentage of micro, meso, and macrospores and I suggest to add only mesopore volume in cm3/g determined by BJH method.
3) In Table 1, BET should be given as an integer, without decimal point also in main text BET should be given as an in integer.
4) Peaks on survey XPS spectra (Figures 6a and 6b) should contain descriptions with determined elements.
5) Authors should carefully read the whole text, I was found many editorial mistakes, e.g., line 122 nitrogen should start with a small letter, lines 218, 222, 235, 256, in CO2, 2 should be in subscript. Also, spaces are missing in many places in the manuscript and supplementary information.
6) In the description for Figure 9 and Figure 10a is the lack of method used for determination of pore size distribution.
7) In the section results and discussion the obtained results for Li-S batteries should be compared with properties of Li-S batteries prepared with carbon nanofibers and described by other Authors.
Round 2
Reviewer 2 Report
I accept revised version for publication.